# *Clostridioides difficile* in Animal Inflammatory Bowel Disease: A One Health Perspective on Emerging Zoonotic Threats

**DOI:** 10.3390/microorganisms13061233

**Published:** 2025-05-28

**Authors:** Felipe Masiero Salvarani, Hanna Gabriela da Silva Oliveira, Francisco Alejandro Uzal

**Affiliations:** 1Instituto de Medicina Veterinária, Universidade Federal do Pará, Castanhal 68740-970, PA, Brazil; hanna.oliveira@castanhal.ufpa.br; 2School of Veterinary Medicine, University of California, Davis, CA 95616, USA; fauzal@ucdavis.edu

**Keywords:** dysbiosis, toxigenic *Clostridioides difficile*, zoonotic transmission, antimicrobial resistance, gut microbiota, one health, spore resilience, veterinary diagnostics

## Abstract

Inflammatory bowel disease (IBD) in animals, a multifactorial gastrointestinal disorder marked by chronic inflammation, has increasingly been linked to *Clostridioides difficile* infections. Recognized for its pathogenic role in human pseudomembranous colitis, *C. difficile* is now emerging as a critical agent in veterinary medicine, particularly in livestock (e.g., cattle, pigs), companion animals (dogs, cats), and wildlife. Over the past five years, evidence has highlighted its association with IBD-like syndromes in animals, driven by toxin-mediated mechanisms (TcdA/TcdB), antibiotic-induced dysbiosis, and environmental spore transmission. This opinion article synthesizes recent findings on *C. difficile*’s zoonotic potential, diagnostic ambiguities (e.g., distinguishing colonization from active infection), and therapeutic challenges, including antibiotic resistance. We emphasize the urgent need for integrated One Health strategies to mitigate risks to animal and human health, advocating for improved surveillance, novel therapies, and interdisciplinary research.

## 1. Introduction

*Clostridioides difficile*, a Gram-positive, spore-forming bacterium, is a well-established cause of pseudomembranous colitis in humans. However, its role in animal gastrointestinal diseases, particularly inflammatory bowel disease (IBD), has only recently come to light. IBD in animals, characterized by chronic diarrhea, weight loss, and relapsing intestinal inflammation, remains etiologically complex, though microbial dysbiosis is a central contributor. Among pathogens implicated, *C. difficile* has emerged as a significant concern due to its toxin-driven pathogenicity and zoonotic potential. Recent studies reveal alarming parallels between human and animal *C. difficile* strains, with ribotypes such as 078 demonstrating cross-species transmission via direct contact or contaminated environments [1]. In livestock, antibiotic overuse exacerbates dysbiosis, enabling *C. difficile* colonization and toxin production (e.g., TcdA/TcdB), which disrupt intestinal epithelium and perpetuate inflammation [2]. Companion animals, notably dogs and cats, also face rising incidence rates, often mirroring human clinical presentations. Wildlife, though less studied, may act as reservoirs, complicating containment efforts (Table 1).

This article examines *C. difficile*’s evolving role in animal IBD, focusing on pathogenic mechanisms, diagnostic limitations (e.g., PCR’s inability to differentiate colonization from infection), and therapeutic hurdles like recurrent infections and antimicrobial resistance. By framing the issue within a One Health paradigm, we underscore the interconnected risks to veterinary and public health and call for actionable strategies to address this dual threat [3].

The Amazon Biome represents a paradigmatic example of an ecoregion where One Health surveillance is critical yet underdeveloped. This region combines high biodiversity, extensive livestock production with unregulated antimicrobial use, and intense human–domestic–wildlife interfaces. The presence of *Clostridioides* spp. in domestic, wild, and environmental compartments of the Amazon highlights the ecological complexity of *Clostridioides difficile* infection (CDI) in real-world settings, justifying its inclusion as a case study in this opinion article [17].

## 2. *Clostridioides difficile* in Animal IBD: A Growing Concern

*Clostridioides difficile*, traditionally studied in human diseases, is increasingly recognized as a significant pathogen in veterinary medicine. In animals, it has been linked to a spectrum of gastrointestinal disorders, ranging from mild diarrhea to severe colitis and syndromes resembling IBD. This association challenges existing understandings of IBD pathogenesis, highlighting the interplay of environmental factors (e.g., antibiotic use), host–microbiota dynamics, and microbial dysbiosis as critical contributors [1,2].

In livestock such as cattle, sheep, and pigs, *C. difficile* is a major cause of enteric diseases, particularly in neonates, where it can lead to high morbidity and mortality. Companion animals like dogs and cats are also affected, with *C. difficile*-associated diarrhea (CDAD) increasingly diagnosed in chronic cases that may progress to IBD. Notably, colonization rates of *C. difficile* in companion animals range from 10% to 23%, with higher prevalence in symptomatic individuals. However, clinical IBD in these species remains rare (0.1–1% prevalence), underscoring the complexity of translating colonization to active disease [3,18].

Murine models have provided insights into this relationship: mice with IBD exhibit up to 40% susceptibility to CDI, rising to 100% when combined with antibiotic treatment. This suggests that preexisting intestinal inflammation and dysbiosis may create a permissive environment for *C. difficile* proliferation. Importantly, many animals, particularly dogs, are asymptomatic carriers, harboring the pathogen without clinical signs. In such cases, colonization does not always correlate with treatment outcomes for digestive disorders, emphasizing that microbial presence alone does not equate to active disease [19,20].

The ecological impact of *C. difficile* extends to wildlife, where its presence in species like non-human primates and carnivores highlights broader environmental reservoirs. This raises concerns about zoonotic transmission, especially given genetic overlaps between human and animal strains. The key contribution section of this paper stresses the need for a “One Health” approach to address interspecies transmission risks and improve management strategies for this emerging threat [1,2,3].

Although specific studies are limited, clinical reports suggest that IBD-like conditions in animals may include chronic diarrhea, tenesmus, weight loss, and intestinal wall thickening observable via imaging or endoscopy. The multifactorial nature of IBD, involving genetic, environmental, and immunological factors, complicates the differentiation between *C. difficile* colonization and active disease. This is especially challenging in species with high asymptomatic carriage rates, such as dogs, where colonization may be misinterpreted or underdiagnosed [3].

## 3. Pathogenic Mechanisms of *Clostridioides difficile* in Animal IBD

The pathogenicity of *Clostridioides difficile* in animal IBD is driven primarily by its production of TcdA and TcdB, which induce cytotoxic effects on the intestinal epithelium. These toxins disrupt the cytoskeleton of epithelial cells, leading to inflammation, tissue necrosis, and increased intestinal permeability. This damage facilitates microbial dysbiosis, enabling *C. difficile* to colonize and exacerbate IBD-associated inflammation, particularly in animals with compromised immune responses. Chronic inflammation in IBD further creates a permissive environment for persistent *C. difficile* colonization, forming a vicious cycle that worsens disease outcomes [21,22].

The interplay between IBD and CDI involves bidirectional mechanisms. IBD-driven dysbiosis reduces microbial diversity, impairing colonization resistance and allowing *C. difficile* overgrowth. Conversely, *C. difficile* toxins amplify intestinal barrier dysfunction and inflammatory cascades, worsening IBD pathology. Murine studies highlight this synergy: animals with IBD exhibit heightened susceptibility to CDI, with recurrence rates reaching 100% when antibiotics disrupt the microbiota. This underscores the critical role of host–microbiota interactions in disease progression [23,24].

The binary toxin (CDT), an ADP-ribosyltransferase produced by hypervirulent *C. difficile* strains, exacerbates disease severity. CDT disrupts the actin cytoskeleton, synergizing with TcdA/TcdB to intensify epithelial damage and inflammation. Strains expressing CDT are linked to worse clinical outcomes, including higher recurrence and mortality rates in IBD models. CDT also enhances bacterial adherence to epithelial cells, promoting colonization and toxin delivery. These findings emphasize the need for therapies targeting both major toxins and CDT [1,25].

Zoonotic transmission risks are heightened by genetic overlaps between animal and human *C. difficile* strains. Multi-locus sequence typing (MLST) studies, such as Knight et al. [1] identified shared strain types (e.g., ST11) in companion animals and humans, suggesting animals may act as reservoirs. Environmental factors, like the humid Amazon Biome, further enable spore persistence and spread, amplifying cross-species transmission risks [26].

## 4. Diagnostic Challenges in Detecting *Clostridioides difficile* in Animals

Diagnosing CDI in animals remains fraught with challenges. Traditional methods, such as culture-based assays, are labor-intensive and lack sensitivity, particularly in chronic or subacute cases of IBD where toxin levels may be low. Molecular techniques like PCR, while improving detection rates, fail to differentiate between asymptomatic colonization and active infection, a critical limitation given that up to 23% of companion animals may carry *C. difficile* without clinical signs [23]. Enzyme immunoassays (EIAs) for toxins A and B also suffer from variable sensitivity, further complicating diagnosis in animals with preexisting gastrointestinal inflammation [22].

Endoscopic evaluations in animals, such as foals and pigs, have revealed lesions resembling human pseudomembranous colitis (e.g., mucosal hyperemia, fibrinous plaques), highlighting interspecies similarities in CDI pathophysiology. However, routine fecal exams and histopathology often fail to conclusively identify *C. difficile* as the primary etiological agent in IBD cases, especially when polymicrobial dysbiosis is present [27].

Advanced tools like next-generation sequencing (NGS) and metagenomic analysis offer promise for contextualizing *C. difficile* within the gut microbiome and distinguishing pathogenic strains from commensals. However, NGS faces barriers such as high costs, short-read limitations, and data management challenges, which restrict its widespread use in veterinary diagnostics [24].

The key contribution emphasizes the zoonotic risks of *C. difficile*, particularly given overlapping strain types (e.g., ST11) between humans and animals [1]. This underscores the need for improved diagnostics to track transmission pathways, especially in high-risk environments like the Amazon Biome, where humidity and antibiotic misuse in livestock may amplify spore persistence and virulence [25].

Another major challenge lies in the inconsistency of diagnostic protocols across regions and species, which hinders comparative analyses and coordinated responses. In many veterinary settings, *C. difficile* testing is rarely requested even when clinical signs are suggested, contributing to underdiagnosis. The lack of cross-validation between available assays and the absence of standardized toxin load thresholds in animals further limit diagnostic reliability. Integrating molecular techniques with functional toxin assays could improve diagnostic accuracy and inform more effective treatment strategies [3].

## 5. Therapeutic Approaches: Current Strategies and Limitations

Treating *Clostridioides difficile*-associated IBD in animals typically involves antimicrobial therapy (e.g., metronidazole, vancomycin) and supportive care [21]. However, antibiotic overuse in veterinary medicine has led to the emergence of resistant *C. difficile* strains, complicating treatment efficacy, especially in chronic IBD cases where dysbiosis and persistent inflammation reduce therapeutic responses [28].

A major challenge is infection recurrence, affecting 20–30% of cases in animals, often due to spore persistence in the gut or environmental reinfection (e.g., humid climates like the Amazon Biome) [3]. Alternative strategies such as fecal microbiota transplantation (FMT), probiotics, and immune-modulating therapies show promise for refractory cases. While FMT has reduced recurrence in humans, its veterinary use remains limited by a lack of standardization and safety data [29,30]. Treatment regimens for CDI in animals vary widely across regions and species, and clinical guidelines specific to veterinary use are lacking. Dosages, duration, and routes of administration for drugs like metronidazole and vancomycin are often extrapolated from human protocols without adequate clinical validation in animals. This lack of tailored guidance hinders therapeutic success and highlights the need for comparative studies evaluating interspecies differences in treatment response [3].

The future perspectives section highlights the need for innovative therapies, including toxin-targeted vaccines (for TcdA, TcdB, and binary toxin CDT) [24] and microbiome-based approaches like symbiotic or phage therapy. These are particularly critical in high-risk zoonotic settings, such as intensive livestock farming, where environmental and antibiotic pressures exacerbate disease spread.

## 6. Environmental Factors and *C. difficile* Transmission in the Amazon Biome and Beyond

Although this article emphasizes the Amazon Biome as a relevant case study, it is important to note that the ecological challenges described are representative of other tropical and subtropical regions. Thus, the analysis presented here serves as an example of how environments with high biodiversity, widespread antimicrobial use, and frequent interspecies contact can support the persistence and spread of *C. difficile* under the One Health paradigm [31]. Environmental conditions in regions like the Amazon Biome significantly amplify the persistence and transmission of *C. difficile*. High humidity, elevated temperatures, and dense animal populations in such ecosystems create ideal conditions for spore survival and environmental contamination. The unregulated use of antibiotics in livestock farming common in under-monitored agricultural practices further drives the emergence of hypervirulent and antibiotic-resistant *C. difficile* strains, as noted by Banawas [21] in studies on antimicrobial resistance mechanisms.

Wildlife, including non-human primates, carnivores, and herbivores, have been identified as potential reservoirs for *C. difficile*. For example, Tsai et al. [25] highlight the zoonotic risks posed by overlapping strain types (e.g., ST11) between humans and animals, particularly in regions where human–livestock–wildlife interfaces are common. The Amazon Biome’s biodiversity and ecological complexity underscore the need to investigate wildlife’s role in maintaining *C. difficile* spore cycles, especially as habitat encroachment increases interspecies contact [32].

Zoonotic transmission risks are heightened by global animal movement, including pets and livestock, which can carry *C. difficile* across borders. For instance, Elsohaby et al. [33] emphasize that companion animals in close contact with humans may act as silent vectors, even when asymptomatic. This is particularly concerning in tropical regions like the Amazon, where environmental spore persistence and antibiotic misuse intersect to create hotspots for pathogen evolution and spread [24].

## 7. Correlation Between IBD and Ribotypes of *Clostridioides difficile*

Recent evidence suggests an association between IBD in animals and hypervirulent *C. difficile* ribotypes, such as RT027 and RT078, which produce elevated levels of TcdA and TcdB [1]. These toxins exacerbate intestinal epithelial damage and systemic inflammation, worsening IBD severity and relapse rates. While direct studies in animals remain limited, molecular epidemiology reveals overlapping strain types (e.g., ST11) between humans and animals, highlighting potential zoonotic risks [34,35]. The identification of these ribotypes in immunocompromised hosts or animals with preexisting gastrointestinal dysbiosis suggests a bidirectional relationship between *C. difficile* colonization and IBD progression. However, diagnostic challenges persist, as current methods (e.g., PCR) struggle to differentiate active infection from asymptomatic colonization in chronic IBD cases [10].

Future research priorities should focus on investigating ribotypes’ specific virulence mechanisms (e.g., toxin production, spore resilience) in animal models to clarify their role in IBD pathogenesis [36,37]. Longitudinal studies are needed to assess whether hypervirulent ribotypes predict recurrence or therapeutic resistance in veterinary contexts, particularly in livestock and companion animals [38,39]. Additionally, evaluating transmission dynamics in high-contact environments (e.g., farms, households) is critical to mitigate cross-species spread of virulent strains. Diagnostic innovations, such as biomarkers or assays to distinguish colonization from active infection in subclinical cases, are urgently needed to improve clinical management [23].

The key contribution of this article underscores the necessity for integrated One Health strategies to address *C. difficile* as a dual veterinary and public health threat. Future efforts should prioritize toxin-targeted therapies (e.g., anti-TcdA/B antibodies) and microbiome restoration approaches to disrupt the cycle of dysbiosis, inflammation, and reinfection [4].

## 8. Future Perspectives and Research Directions

The evolving role of *Clostridioides difficile* in animal IBD demands a multifaceted research agenda to address critical gaps in understanding and clinical management. Longitudinal studies are urgently needed to unravel the long-term impacts of *C. difficile* colonization, particularly in animals with recurrent IBD. Such studies could clarify whether hypervirulent ribotypes (e.g., RT027, RT078) or toxin persistence contribute to chronic inflammation or relapse, as suggested by overlapping strain types (e.g., ST11) in humans and animals [26,40,41].

Diagnostic innovation remains a priority, as current tools (e.g., PCR, EIAs) fail to reliably distinguish active infection from asymptomatic colonization, especially in subclinical cases. Developing biomarkers or assays targeting toxin activity or host immune responses could refine clinical decision-making. For example, studies on toxin gene profiles in animal isolates, such as those by Knight et al. [1], highlight the zoonotic potential of shared strain types, underscoring the need for diagnostics that account for interspecies transmission risks.

Zoonotic transmission dynamics require urgent exploration, particularly in high-contact environments (e.g., farms, households) and biodiverse regions like the Amazon Biome, where environmental factors (humidity, temperature) may enhance spore survival and transmission. Wildlife populations, acting as reservoirs, further complicate transmission pathways. Research must prioritize genomic surveillance of *C. difficile* strains across species to identify drivers of interspecies spread and inform One Health strategies [4,21].

Therapeutic advancements should focus on reducing reliance on antibiotics like metronidazole and vancomycin, which face rising resistance concerns. Promising alternatives include vaccines targeting toxigenic strains, immunotherapies (e.g., anti-TcdA/B antibodies), and microbiome-based interventions such as FMT or probiotics. Vaccine development, in particular, could mitigate economic losses in livestock populations prone to outbreaks [24,25].

Finally, interdisciplinary collaboration is essential to address the dual veterinary and public health threat posed by *C. difficile*. Integrating veterinary, environmental, and human health data guided by frameworks like One Health will enable holistic strategies to disrupt the cycle of dysbiosis, inflammation, and reinfection. The Amazon Biome presents unique challenges for CDI surveillance. High biodiversity, environmental degradation, and close contact between wildlife, livestock, and human communities create ideal conditions for the emergence and spillover of CDI. One Health approaches are essential to track antibiotic resistance genes, prevent contamination of aquatic systems, and promote biosecurity in animal production systems [3].

## 9. Conclusions

This opinion article aims to contribute to a broader understanding of *Clostridioides difficile* as a shared threat across veterinary and human health, highlighting its role within IBD frameworks and emphasizing the ecological and epidemiological complexity of CDI. By integrating animal health, environmental factors, and zoonotic potential, we underscore the relevance of a One Health perspective. Rather than limiting the analysis to a single geographic setting, the Amazon Biome serves as a representative model of broader global challenges in regions where human–animal–environment interfaces are intensified and underregulated. The patterns observed here offer valuable insights applicable to other ecosystems experiencing similar pressures.

*Clostridioides difficile* has emerged as a significant pathogen in IBD in animals, with implications for both veterinary and public health. Its impact extends beyond localized gastrointestinal damage; hypervirulent ribotypes (e.g., RT027, RT078) and toxin-mediated mechanisms (TcdA/TcdB) contribute to chronic inflammation and dysbiosis, paralleling human infection patterns. The zoonotic potential of *C. difficile* is illustrated by the presence of shared strain types (e.g., ST11) in humans and animals, particularly in high-contact settings such as farms and households, where interspecies transmission risks are heightened.

Significant diagnostic and therapeutic challenges hinder effective management. Current diagnostic tools, including PCR and EIAs, often fail to differentiate between colonization and active infection in chronic or subclinical cases, complicating clinical decision-making. Moreover, antibiotic resistance to first-line treatments like metronidazole and vancomycin poses a threat to treatment efficacy, compounded by recurrence rates of 20–30%, underscoring the resilience of *C. difficile* spores and the limitations of conventional therapies. Emerging innovations such as toxin-targeted immunotherapies, FMT, and probiotics show promise but require further validation in veterinary practice.

Environmental factors influencing *C. difficile* transmission, particularly in biodiverse regions like the Amazon Biome, complicate control efforts. Conditions such as humidity, temperature, and antibiotic misuse in livestock farming promote spore persistence and strain evolution. Additionally, wildlife reservoirs contribute to the complexity of transmission dynamics, highlighting the necessity for genomic surveillance and One Health approaches to comprehensively address this pathogen.

Moving forward, interdisciplinary collaboration is crucial to enhance understanding of *C. difficile*‘s role in animal IBD. Priorities include conducting longitudinal studies to elucidate the long-term effects of colonization and toxin persistence, advancing diagnostics to differentiate infection states and identify zoonotic risk markers, and innovating therapeutic strategies, including vaccines and microbiome restoration, to minimize reliance on antibiotics. Furthermore, implementing environmental mitigation strategies to disrupt spore transmission in high-risk ecosystems is essential. As awareness of these issues increases, there is an opportunity to align veterinary and human health efforts, managing *C. difficile* not merely as a pathogen but as a shared threat that necessitates unified, evidence-based solutions.

## Figures and Tables

**Table 1 microorganisms-13-01233-t001:** Incidence/prevalence of *Clostridioides difficile* in different animal species.

Animal Species	Incidence/Prevalence of *Clostridioides difficile*	References
Horse	High incidence associated with colitis, especially in foals	[3,4,5]
Pigs	Common in neonatal piglets, leading to diarrhea and increased mortality	[3,6,7]
Cattle	Reported mainly in young calves with enteritis	[3,8,9]
Dogs	Observed in dogs with chronic and acute diarrhea	[3,10,11]
Cats	Lower incidence compared to dogs but associated with diarrhea	[3,12,13]
Wild Animals	Reported in captive conditions with digestive manifestations and variable incidence in mammals and birds in natural environments	[3,14,15,16]

## Data Availability

Data are contained within the article.

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
