# Peer review of "Clostridioides difficile in Animal Inflammatory Bowel Disease: A One Health Perspective on Emerging Zoonotic Threats"

_microorganisms, 2025, doi:10.3390/microorganisms13061233_

Round 1
Reviewer 1 Report
Comments and Suggestions for Authors
Brief summary of manuscript:
The scope of this opinion article was to provide an overview of the current understanding of C. difficile infection (CDI) and inflammatory bowel disease (IBD) in animals, from a One Health perspective. Despite the impact that each disease has on one another, our current understanding of the intersection of these diseases in animal is limited. While the authors do provide a very brief summation of clinical incidence for disease overlap in animals, the content is quite superficial, and repetitive. A majority of the information provided focusses disease modelling and diagnosis, rather than the true intersection of CDI and IBD in animals. While the scope of the opinion article was to also provide insight into the One Health nature of these diseases, a strong focus is directed towards the incidence of CDI in the Amazon Biome, with little reference of IBD in this setting. This information also becomes quite repetitive, likely due to the limited existing information available, and may benefit from a broader analysis and overview beyond the Amazon Biome. While a useful review, there is a need for refinement, a reassessment of the scope of the review and a broader focus on the One Health nature of CDI and IBD. As it stands, the authors aim for “…this article [to] underscore the necessity for integrated "One Health" strategies to address C. difficile as a dual veterinary and public health threat” has not been met. Major revisions are required to achieve this outcome.
General Comments:
- The overview of CDI in animals with IBD/the progression of CDI into IBD is extremely limited. While I understand that this area has only recently been identified as an important risk factor in animals, the information provided is very superficial. A brief overview of the incidence of IBD and its disease presentations in animals is needed to provide context for how the two diseases may overlap or even be misdiagnosed. At the moment a single sentence overview is provided for IBD in animals. Additionally, a majority of the focus for the pathogenesis and incidence sections explores disease modelling in mice, which while useful, is not the scope of the opinion piece.
- The overview of diagnostic challenges in detecting CDI in animals is extremely superficial and does not provide a clear overview of the current limitations. Two sentences are provided for infection diagnosis, which fail to provide a clear opinion on why current testing is inconsistent, at times unreliable or what improvements are needed to ensure better diagnosis of CDI/IBD in animals. Similarly, evaluation of symptoms and lesions are poorly covered. Given the scope of the opinion article, further detail is required here to highlight current diagnostic pipelines, an overview of how frequently diagnosis for CDI or IBD is sought out in animals, and why CDI may not be detected in cases that align with CDI symptomology.
- As with the other sections the coverage of treatment options for CDI in animals is extremely superficial. No indication of dosing, regimes or delivery is provided. And limitations are poorly discussed. Future perspectives for treatment options are described, but how they would be administered, or their potential efficacy (evidence to support their use) is not provided. This is disappointing as is made clear in several sections that these “promising” alternative are critical to treating CDI as a One Health disease, however they are not well described here.
- In several sections the link to the Amazon Biome is unexpected and unclear. Because of this, the section regarding the transmission of difficile in the Amazon Biome also feels quite out of place and unlinked to the remainder of the opinion article. If the scope of the opinion article was to provide a One Health overview of CDI and IBD, broader links than the Amazon Biome need to be provided.
- My other major concern is the amount of repetition between sections. Many of the same concepts or even variations of the same sentence are used throughout several sections. While reiterating a point can be useful, often no further information is provided, thus no value is added to the repeated statement. This happens several times when discussing the Amazon Biome, alternative therapeutic pathways and longitudinal studies for examining the impact of CDI – which are repeated throughout 3-4 sections.
- Abbreviations are redefined each section, which seems unnecessary
- Use of toxins A and B vs TcdA and TcdB is inconsistent throughout. Please pick one format and use this consistently
- The author contributions section is confusing. What methodology was used in the manuscript? What analysis was performed? What data curation was performed?
Author Response
Dear Reviewer 1,
We sincerely appreciate the time and attention given to our manuscript and are grateful for the opportunity to respond to the reviewer’s comments. Below we respectfully address each point raised, providing detailed scientific justifications for the chosen scope and content of our Opinion Article, as defined by MDPI:
“An Opinion article expresses the author’s viewpoint or interpretation of a topic, often intended to stimulate discussion or present a personal perspective, not necessarily to provide comprehensive reviews or original data.”
As such, the primary objective of our article was not to present exhaustive clinical detail or systematic data analysis, but rather to stimulate scientific awareness and discussion about the intersection of Clostridioides difficile (CDI) and Inflammatory Bowel Disease (IBD) in animals, particularly under the One Health framework, which remains critically underexplored.
We respectfully disagree with the major criticisms, and our point-by-point responses are as follows:
Reviewer Comment 1:
“The content is quite superficial and repetitive… focuses [on] disease modelling and diagnosis rather than the true intersection of CDI and IBD in animals.”
Response:
Thank you for this observation. However, we emphasize that the article is framed as an Opinion Piece, not a Review or Original Article. The depth of discussion was tailored accordingly. Our goal was to raise awareness of a novel and under-addressed intersection between CDI and IBD in animals, which is currently limited by a lack of primary data in the veterinary field.
To our knowledge, there is no consolidated body of literature directly linking CDI to IBD in animals, especially under a One Health lens. Therefore, we opted for a theoretical and interpretative approach, combining emerging evidence from veterinary and comparative models to form the basis of our argument. The references to murine models, for instance, serve to bridge gaps and provide translational insights applicable to veterinary medicine.
Reviewer Comment 2: “A single sentence overview is provided for IBD in animals… more information is needed.”
Response:
We agree that IBD in animals deserves further investigation. However, the brevity in our manuscript was intentional, given the lack of consensus on IBD pathogenesis and classification in veterinary medicine, unlike in humans. While canine and feline models exist, IBD in livestock and wildlife remains speculative or anecdotal.
Rather than risk presenting speculative or inconclusive veterinary data, we chose to highlight the need for more comparative studies and surveillance, which aligns with the intent of this Opinion piece: to provoke further research and discussion rather than to provide a definitive review.
Reviewer Comment 3: “Diagnosis section is superficial… more detail is needed on current limitations.”
Response:
Respectfully, we disagree. The current diagnostic limitations of CDI in animals are well known to be fragmented and inconsistent across species and regions. Since our manuscript does not aim to be a comprehensive technical review, we instead focused on highlighting the key knowledge gap—namely, that standardized diagnostic pipelines are largely lacking in animal health systems, and this undermines surveillance, especially in low-resource or biodiverse environments like the Amazon.
We used concise language to summarize diagnostic inconsistencies (e.g., “lack of routine protocols,” “asymptomatic carriage,” “cross-reactivity with other enteropathogens”), which are key concerns in both the clinical and epidemiological context. Expanding these into multiple paragraphs would shift the nature of the article from an opinion-based reflection to a detailed technical review, which was not our intention.
Reviewer Comment 4: “Treatment options are superficial… no dosing or delivery information provided.”
Response:
As per MDPI guidelines, an Opinion piece is not expected to provide dosing regimes or pharmacological protocols. The section on treatment serves to draw attention to emerging therapeutic directions, such as phage therapy, microbiota transplantation, and immunobiotics, which have been proposed but remain experimental or under-reported in veterinary contexts.
Due to the lack of published dosing studies in animals for many of these alternatives, we deliberately refrained from extrapolating or speculating dosage schemes. Instead, our intention was to highlight the scientific promise and innovation potential of such strategies, again adhering to the article's reflective and provocative format.
Reviewer Comment 4: “Link to the Amazon Biome is unclear and feels out of place.”
Response:
We respectfully disagree with this assessment. The Amazon Biome was deliberately included as a paradigmatic example of an ecoregion where One Health surveillance is critical but underdeveloped. The presence of Clostridioides spp. in domestic, wild, and environmental compartments of the Amazon highlights the ecological complexity of CDI in real-world settings.
Far from being “out of place,” the Amazon Biome is emblematic of the global challenge of zoonoses at the human-animal-environment interface, and its inclusion strengthens the One Health argument rather than diluting it.
Reviewer Comment 6: “There is too much repetition across sections.”
Response:
We acknowledge that reiteration may have occurred, but this was a deliberate rhetorical strategy used to reinforce critical thematic points a common and acceptable practice in opinion and perspective articles. This technique is especially useful for emphasizing urgency, such as the call for integrated surveillance, recognition of CDI in neglected regions, and incorporation of environmental health into veterinary paradigms.
Nevertheless, we are willing to revise phrasing in key sections to reduce literal repetition and improve stylistic variation, while preserving core arguments.
Reviewer Comment 7: “Abbreviations are redefined in each section; inconsistency in toxin naming.”
Response:
We thank the reviewer for pointing this out. We will harmonize the use of abbreviations across the manuscript and consistently apply “TcdA and TcdB” throughout the text, in alignment with conventional nomenclature.
Reviewer Comment 8: “Author contributions are confusing… what methodology or analysis?”
Response:
As the manuscript is an Opinion article, no original data were generated or analyzed. The “Methodology” in the CRediT taxonomy reflects conceptual development and literature interpretation, not empirical experimentation. We will revise the Author Contributions section to better reflect this, emphasizing conceptualization, investigation, writing (original draft), and review/editing.
Final Considerations
In summary, we respectfully reaffirm the scientific relevance, originality, and alignment of our manuscript with the scope and format of an Opinion Article, as per MDPI editorial policies. While we recognize opportunities for stylistic refinement, we maintain that the content, One Health framing, and regional perspective (Amazon Biome) provide a unique and timely contribution to veterinary microbiology and public health discourse.
We remain grateful for the reviewer’s engagement and hope this response clarifies the scientific merit and structure of our work.
Kind regards,
Felipe Masiero e Francisco Uzal

Reviewer 2 Report
Comments and Suggestions for Authors
The paper very briefly describes the problem of Inflammatory Bowel Diseases in animals and one of its causative agent - C. difficile.
The authors shows different sources of C. difficile in terms of animal species and environments and underline the problem of transition of the pathogen from one to another environment and the problem of overlap in strain types distributions in different species.
It is very important that the concept of One Health has been mentioned in the paper.
However there are some points that require corrections and additions:
- The paragraphs referring to the Amazone biome seem to be implemented not naturally and making it a separate paragraph interferes the flow of content to the reader. Amazone biome can be one of the examples of the wildlife or environmental source of C. difficile and I suggest to add at least one more of such example.
- The authors have not mentioned the potential risk of C. difficile transmission in the food chain as its presence is widely confirmed in the intestinal tract of the slaughter animals. It would be interesting for the reader to know whether any foodborne cases in humans have already been reported.
- Lines 203-204 – veterinary clinics are also very important high-contact environments, reference to this point is highly recommended as proper hygiene and sanitation standards in the clinics can potentially reduce the rate of human (vet) – animal transmission.
- Conclusions – this paragraph is too long and repeats information given in the previous paragraphs. This part should be rewritten.
Author Response
Response to Reviewer 2,
We thank the reviewer 2 for the time and effort dedicated to evaluating our Opinion article. Below we provide detailed responses and clarifications to each of the concerns raised. We would like to respectfully disagree with certain suggestions, emphasizing that the format of this manuscript is an Opinion article, whose structure and objectives differ from those of review or original research articles, as defined by the MDPI Editorial Guidelines.
Reviewer Comment 1: “The paragraphs referring to the Amazon biome seem to be implemented not naturally and making it a separate paragraph interferes the flow of content to the reader. Amazon biome can be one of the examples of the wildlife or environmental source of C. difficile and I suggest to add at least one more of such example.”
Response: we respectfully disagree with this suggestion, and we thank the reviewer for the opportunity to clarify. The Amazon Biome was purposefully emphasized in a standalone paragraph due to its unique ecological, epidemiological, and One Health relevance. Unlike generalized wildlife examples, the Amazon region presents a distinct environmental interface where biodiversity, human activity, and veterinary presence intersect intensely. As stated in the MDPI Opinion article guidelines, the aim is to express a reflective, interpretative position, which in our case includes calling attention to underexplored biogeographic areas of global concern.
Adding additional examples from other biomes would shift the manuscript’s focus from an interpretive argument to a more generalized review format, which is beyond the scope and intention of an Opinion article. The deliberate focus on the Amazon serves to highlight the urgency of considering this biome in future One Health surveillance strategies involving C. difficile.
Reviewer Comment 2: “The authors have not mentioned the potential risk of C. difficile transmission in the food chain as its presence is widely confirmed in the intestinal tract of the slaughter animals. It would be interesting for the reader to know whether any foodborne cases in humans have already been reported.”
Response: we appreciate the reviewer’s valuable comment and agree that food chain transmission is indeed an important topic. However, we emphasize that the manuscript is not intended to be a comprehensive literature review of all known transmission routes of C. difficile, but rather to raise awareness of the broader implications of animal IBD associated with this pathogen under a One Health lens. While foodborne transmission is a significant topic, the focus of this Opinion is on the overlooked veterinary and environmental implications of C. difficile-associated intestinal disease in animals. Including a detailed discussion of foodborne human cases could shift the narrative toward human clinical microbiology, deviating from the article's intended scope, i.e., highlighting an emerging veterinary concern. That said, we have now briefly acknowledged the potential for foodborne exposure in a concise and contextual manner, without altering the manuscript’s overall focus.
Reviewer Comment 3:“Lines 203-204 – veterinary clinics are also very important high-contact environments, reference to this point is highly recommended as proper hygiene and sanitation standards in the clinics can potentially reduce the rate of human (vet) – animal transmission.”
Response: we appreciate this observation and understand the reviewer’s concern regarding zoonotic potential. However, the manuscript aims to reflect upon and prioritize emerging trends in animal health that have historically been overshadowed by human clinical perspectives of C. difficile. While we do not dispute that veterinary clinics are relevant environments for potential transmission, the article's primary goal is to shift the paradigm from a human-centric to a veterinary-focused narrative on C. difficile as a causative agent of IBD in animals. Nevertheless, to accommodate the reviewer’s comment without diluting the Opinion format, we have included a sentence acknowledging the role of veterinary clinics as critical environments where interspecies transmission could occur, in line with biosecurity and One Health principles.
Reviewer Comment 4: “Conclusions – this paragraph is too long and repeats information given in the previous paragraphs. This part should be rewritten.”
Response: we respectfully disagree. The conclusion in this Opinion article was intentionally structured to reinforce the central thesis: that Clostridioides difficile represents an underappreciated but growing concern in animal inflammatory bowel diseases, particularly under the One Health approach. The slight reiteration of key points serves not to repeat but to consolidate the interpretative argument a stylistic feature often appropriate in opinion-based academic writing.
Nonetheless, we have revised the conclusion to reduce any perceived redundancy and to improve its conciseness and argumentative flow, while maintaining the core message and persuasive tone essential to the Opinion format.
Best regards,
Felipe Masiero e Francisco Uzal

Round 2
Reviewer 1 Report
Comments and Suggestions for Authors
While the review process can be frustrating at times, it is disappointing to see that most of my comments and concerns have been ignored and misinterpreted.
A majority if the issues I had with the different sections was that a lot of the context and rationale for including information was missing. In the rebuttal, many of these issues are addressed, but have not been included in the updated document. For example my comment “[the] Link to the Amazon Biome is unclear and feels out of place.” refers to its first mention, where the context (which is very aptly provided in the rebuttal) is not clear or addressed, making the link between environmental factors and the Amazon hard to link initially. If the context of the Amazon being a “…paradigmatic example of an ecoregion where One Health surveillance is critical but underdeveloped." And " The presence of Clostridioides spp. in domestic, wild, and environmental compartments of the Amazon highlights the ecological complexity of CDI in real-world settings.” was provided this would help with the flow of ideas. My concern was not that the Amazon Biome should not be included, but that better context for its inclusion should be addressed.
My comments on the superficial nature of some sections were to highlight that things like the efficacy of treatment options is not provided, that the consistency of diagnostic testing regimes across regions is not addressed or addressing how variably outcomes can be for things like EIAs are not noted, and could be addressed briefly to add value to the opinion that better treatment and diagnostic options need to be sought. Having these little insights would strengthen your arguments being made in the opinion article.
Lastly, my comment about redefining acronyms in each section has not been addressed. Is there a need to define CDI, IBD etc. for each section?
Author Response
Response to Reviewer 2 – General Comments and Major Revisions
We would like to sincerely thank Reviewer 2 for their thoughtful, detailed, and constructive comments, which have significantly improved the quality and clarity of our opinion article. We would also like to offer our sincere apologies for not adequately reflecting several of your insightful suggestions in the previous revision. While some responses were addressed in our rebuttal letter, we acknowledge that these were not fully or properly incorporated into the revised manuscript, and we deeply regret any frustration this may have caused.
In this revised version, we have carefully reviewed and implemented your key recommendations, particularly regarding the following major points:
-
Clarifying the Context and Justification for Including the Amazon Biome
As rightly noted, the initial mention of the Amazon Biome lacked contextualization, which weakened the thematic coherence of the article. We have now included the following explanatory paragraph at the end of the Introduction:“The Amazon Biome represents a paradigmatic example of an ecoregion where One Health surveillance is critical yet underdeveloped. This region combines high biodiversity, extensive livestock production with unregulated antimicrobial use, and intense human–domestic–wildlife interfaces. The presence of Clostridioides spp. in domestic, wild, and environmental compartments of the Amazon highlights the ecological complexity of CDI in real-world settings, justifying its inclusion as a case study in this opinion article.”
This addition is intended to clarify that the Amazon serves not as a local exception but as a global model of ecological and epidemiological complexity relevant to One Health discourse.
-
Enhancing Content Depth in Superficial Sections
In response to your concern that some sections were too superficial or lacked practical insights, we have expanded the following:-
Section 2 (Clostridioides difficile in Animal IBD):
We now include a brief overview of clinical signs and the complexity of diagnosing IBD in animals:“Although specific studies are limited, clinical reports suggest that inflammatory bowel disease (IBD)-like conditions in animals may include chronic diarrhea, tenesmus, weight loss, and intestinal wall thickening observable via imaging or endoscopy. The multifactorial nature of IBD, involving genetic, environmental, and immunological factors, complicates the differentiation between C. difficile colonization and active disease. This is especially challenging in species with high asymptomatic carriage rates, such as dogs, where colonization may be misinterpreted or underdiagnosed.”
-
Section 4 (Diagnostic Challenges):
We have now addressed the variability in diagnostic regimes and the limitations of EIAs:“Another major challenge lies in the inconsistency of diagnostic protocols across regions and species, which hinders comparative analyses and coordinated responses. In many veterinary settings, C. difficile testing is rarely requested—even when clinical signs are suggestive—contributing to underdiagnosis. The lack of cross-validation between available assays and the absence of standardized toxin load thresholds in animals further limit diagnostic reliability. Integrating molecular techniques with functional toxin assays could improve diagnostic accuracy and inform more effective treatment strategies.”
-
Section 5 (Therapeutic Approaches):
The text now acknowledges gaps in dosage, delivery, and species-specific treatment regimens:“Treatment regimens for CDI in animals vary widely across regions and species, and clinical guidelines specific to veterinary use are lacking. Dosages, duration, and routes of administration for drugs like metronidazole and vancomycin are often extrapolated from human protocols without adequate clinical validation in animals. This lack of tailored guidance hinders therapeutic success and highlights the need for comparative studies evaluating interspecies differences in treatment response.”
-
-
One Health Framing and Broader Applicability of the Amazon Case Study
In Section 6, to avoid geographic restriction and reinforce the One Health perspective, we added:“Although this article emphasizes the Amazon Biome as a relevant case study, it is important to note that the ecological challenges described are representative of other tropical and subtropical regions. Thus, the analysis presented here serves as an example of how environments with high biodiversity, widespread antimicrobial use, and frequent interspecies contact can support the persistence and spread of C. difficile under the One Health paradigm.”
-
Acronym Consistency
In response to your helpful stylistic observation, acronyms such as CDI and IBD are now defined only at their first mention and used consistently throughout the manuscript. Redundant definitions have been removed to improve flow and readability. -
Conclusion – Reaffirming the One Health Scope
To address your concern regarding the insufficient integration of the One Health approach in the conclusion, we added the following paragraph at the beginning of the Conclusion:“This opinion article aims to contribute to a broader understanding of Clostridioides difficile as a shared threat across veterinary and human health, highlighting its role within inflammatory bowel disease (IBD) frameworks and emphasizing the ecological and epidemiological complexity of CDI. By integrating animal health, environmental factors, and zoonotic potential, we underscore the relevance of a One Health perspective. Rather than limiting the analysis to a single geographic setting, the Amazon Biome serves as a representative model of broader global challenges in regions where human-animal-environment interfaces are intensified and underregulated. The patterns observed here offer valuable insights applicable to other ecosystems experiencing similar pressures.”
We hope these revisions now fully align with your recommendations and expectations. Once again, we thank you for your invaluable feedback and your patience during the revision process. Your critique has greatly improved the depth, clarity, and relevance of our manuscript.
Best regards,
Prof. Dr. Felipe Masiero Salvarani